# Network Memory Footprint Compression Through Jointly Learnable Codebooks and Mappings

**Edouard Yvinec**[1,2], **Arnaud Dapogny**[2], **Kevin Bailly**[1,2]
Sorbonne Université[1], CNRS, ISIR, f-75005, 4 Place Jussieu 75005 Paris, France
Datakalab[2], 114 boulevard Malesherbes, 75017 Paris, France
ey@datakalab.com

## Abstract

The massive interest in deep neural networks (DNNs) for both computer vision and natural language processing has been sparked by the growth in computational power. However, this led to an increase in the memory footprint, to a point where it can be challenging to simply load a model on commodity devices such as mobile phones. To address this limitation, quantization is a favored solution as it maps high precision tensors to a low precision, memory efficient format. In terms of memory footprint reduction, its most effective variants are based on codebooks. These methods, however, suffer from two limitations. First, they either define a single codebook for each tensor, or use a memory-expensive mapping to multiple codebooks. Second, gradient descent optimization of the mapping favors jumps toward extreme values, hence not defining a proximal search. In this work, we propose to address these two limitations. First, we initially group similarly distributed neurons and leverage the re-ordered structure to either apply different scale factors to the different groups, or map weights that fall in these groups to several codebooks, without any mapping overhead. Second, stemming from this initialization, we propose a joint learning of the codebook and weight mappings that bears similarities with recent gradient-based post-training quantization techniques. Third, drawing estimation from straight-through estimation techniques, we introduce a novel gradient update definition to enable a proximal search of the codebooks and their mappings. The proposed jointly learnable codebooks and mappings (JLCM) method allows a very efficient approximation of any DNN: as such, a Llama 7B can be compressed down to 2Go and loaded on 5-year-old smartphones.

## 1 Introduction

Deep neural networks (DNNs) have reached a point of hegemony in most areas of machine learning. This is most blatant in the context of computer vision Ren et al. (2015); Chen et al. (2017) and natural language processing Devlin et al. (2018). In particular, the transformer architectures Vaswani et al. (2017) have achieved the most impressive results in terms of performance and predictive relevance, with large language models Zhang et al. (2022a) in the lead. However, these deep architectures and their performance come at a price: their memory footprint. This cost has reached new heights with some architectures requiring multiple modern, high-end, devices to simply load such models. For example, to this day, the largest consumer grade graphics processing unit (GPU) has 24Go of VRAM, which is not enough to load even the smallest member of the Llama LLM family Touvron et al. (2023) using the default floating point (fp32) format (28Go). This computational cost growth has been more and more hindering DNNs integration and deployment, especially on mobile devices.

In practice, this limitation is often addressed using weight quantization techniques. The latter consists in mapping the original high-precision tensors to low-precision, memory efficient representations. The resulting quantized model requires a significantly lower memory footprint and often leverages less costly individual operations. However, the quantization process itself may be costly, as the best performing such solutions (namely QAT, short for Quantization-Aware Training Zhang et al. (2022b)) usually involves full retraining, sometimes with larger batch sizes and number of

Standard Backpropagation

| Hard Indices | | | Soft Index | Codebook | Index Gradient Update |
|---|---|---|---|---|---|
| 0 | 3 | 1 | | -1.23 | |
| 0 | 2 | 1 | | -0.09 | |
| 2 | 0 | 2 | | 0.17 | |
| 3 | 1 | 0 | | 0.96 | |

JLCM (multiple codebooks)

| Hard Indices | | | Soft Index | Codebooks | Index Gradient Update |
|---|---|---|---|---|---|
| 0 | 3 | 1 | | -1.23 | |
| 0 | 2 | 1 | | -0.09 | |
| 2 | 0 | 2 | | 0.17 | |
| 3 | 1 | 0 | | 0.96 | |

(additional codebook: -2.01, 0.13, 0.77, 1.52)

JLCM (multiple scales)

| Hard Indices | | | Soft Index | Codebook | Scales | Index Gradient Update |
|---|---|---|---|---|---|---|
| 0 | 3 | 1 | | -1.23 | 1.02 | |
| 0 | 2 | 1 | | -0.09 | 0.89 | |
| 2 | 0 | 2 | | 0.17 | 0.97 | |
| 3 | 1 | 0 | | 0.96 | 1.12 | |

Figure 1: Left side: in the standard approach, each weight is represented by a soft index that points to a value within a single codebook. Assuming that the loss for this weight suggests increasing its value (highlighted in red), then the standard gradient update would be scaled by the codebook values, thus skipping 0.17 and directly increasing the weight of 0.96 in the codebook. Center/right side: *a contrario*, JLCM leverages multiple codebooks and/or scales and defines a novel gradient update term to allow proximal search within the codebook values.

epochs. To circumvent this and scale to any DNN size, post-training quantization has been introduced, with the most extreme scenario being the fully data-free setup, as described in Nagel et al. (2019), however generally at a significant expanse in terms of accuracy of the quantized models. Noteworthy, beyond this fully data-free setting, the performance of quantization techniques can be greatly overhauled by considering very small calibration sets, as in the work of Nagel et al. (2020).

The primary aim of the aforementioned quantization techniques is to reduce the latency of the network by performing arithmetic in lower precision formats. This may however introduce unnecessary constraints if the goal is to rather limit the memory footprint. In order to do this, the most effective methods in terms of compression rates consists in storing in a codebook a restricted set of values, onto which each weight shall be mapped using a more compact code. Nevertheless, these methods suffer from two major drawbacks , as illustrated on Figure 1. First, the *granularity* problem: these approaches use a single codebook for each tensor in order to avoid a more costly, conditional mapping: intuitively, if multiple codebooks were to be used, one would need to store both the codebook and value indexes for each weight (*vs.* only the value index in a single codebook), which would dramatically increase the memory footprint. Second, we argue that the optimization of the mapping itself is not well-defined. Formally, for a given scalar weight value, a gradient-based optimization would favor jumps towards extreme values rather than closer values in the target codebook. This would be in opposition with the desired proximal behavior of gradient descent optimization. As a result, codebooks and mappings cannot be learned in a joint manner using gradient-based algorithms, limiting their efficiency.

In this work, we propose to tackle these two limitations, as pinpointed on Figure 1. First, we propose to group similar neurons as defined by their weight distribution. We can then apply different scaling factors to homogeneous weight values or, alternatively, we can assign multiple codebooks to the reordered weight matrix without additional memory overhead (as, in this case, the indexation to different codebooks depends implicitly on the weight index). This provides a finer grained initialization. Second, stemming from that initialization, we propose to jointly optimize the codebooks and mappings in a setup similar to what is done in gradient-based post training quantization. Third, drawing inspiration from straight-through estimation in DNN quantization, we propose to rewrite the gradient update for the codebook mappings, favoring proximal values in the codebooks. The proposed jointly learnable codebooks and mappings (JLCM) method can be summarized in:

- a neuron re-ordering based on weight distribution similarities. This can be leveraged to apply different scale factors on homogeneous weight groups, or map these groups to multiple codebooks without any memory overhead over the mappings.

- a joint learning of the codebooks and mappings that bears similarities with gradient-based post-training quantization techniques.

- a new gradient update definition for the codebook assignment, which favors proximal values over extreme values.

The proposed JLCM method achieves almost ternary compression of all the layers of large transformers with less than one percent performance degradation. Such compression rates are of particular interest for large language models deployment at scale. For instance, JLCM enables us to

preserve the 95% of the performance of a Llama 7B model on a 2Go device such as the iPhone 8 or a Samsung Galaxy A10.

## 2 RELATED WORK

In this section, we delve into the current state-of-the-art in the related DNN quantization techniques, as well as the recent gradient-based post-training quantization techniques. We also review existing codebook-based methods for DNN memory footprint compression.

### 2.1 QUANTIZATION

In post training quantization, the data-free setup was first introduced in Nagel et al. (2019). In this work, the core challenge at hand was to identify quantization ranges for both weights and activations without having access to any data. To do this, the authors proposed to use statistics stored during training by the batch-normalization layers. With respect to the weight tensors, the proposed method leveraged baseline uniform operators Krishnamoorthi (2018) between floating points and integers. Since this work, many iterations Zhao et al. (2019); Cong et al. (2022) have been proposed, focusing on improving the weights encoding. However, all the aforementioned methods struggle to retain a good accuracy when weights are quantized using 4 or fewer bits, even on models that are nowadays deemed less challenging to quantize such as ResNet 50 He et al. (2016). This shortcoming is often addressed with non-uniform quantization. For instance, Li et al. (2019) propose to use the logarithm function to map weight values to an integer exponent: this changes the nature of the arithmetic used by the quantized networks, with multiplications becoming bit shifts. Yvinec et al. (2023c) argue that such change is hard to leverage on existing hardware in practice, and propose a power quantization method that preserve the nature of these operations. However, this method also requires custom implementation to provide latency boosts on existing hardware. Furthermore, on LLMs, fully data-free power quantization does not enable quantization below the 4 bits mark, even combined with group-wise quantization techniques.

### 2.2 GRADIENT-BASED POST-TRAINING QUANTIZATION

In order to circumvent this limitation while maintaining the scalability of data-free quantization, Nagel et al. (2020) introduced AdaRound, the first gradient-based post-training quantization (GPTQ) technique. This method and its iterations Li et al. (2021); Wei et al. (2022); Liu et al. (2023); Yvinec et al. (2023b)consist in optimizing the quantized weights over a small calibration set excerpted from the original training data. AdaRound optimizes each layer individually and sequentially in a self-distillation Hinton et al. (2014) fashion, where the quantized model learns to imitate its original counterpart. This optimization framework has inspired other compression domains, such as pruning Kwon et al. (2022). However, the primary goal of these methods is to reduce the latency of the quantized models: however, if the purpose is to reduce the memory footprint of the models, GPTQ methods struggle to scale to very large networks, e.g. LLMs. In terms of raw memory footprint compression, the most effective methods are the more dedicated codebook-based methods.

### 2.3 CODEBOOK-BASED APPROACHES FOR MEMORY FOOTPRINT REDUCTION

Codebook-based methods Chen et al. (2015); Han et al. (2015); Eban et al. (2020); Jeon et al. (2020); Javaheripi et al. (2022); Yvinec et al. (2021) consist in mapping a continuous set of weight values to a finite set, and each weight is represented on fewer bits by an index referring to one of the codebook values. Among these methods, most focus on effective implementations Chen et al. (2015); Jeon et al. (2020); Javaheripi et al. (2022) and use naive approaches with respect to the mapping step itself. On the flip side, other techniques propose to learn the codebook values Han et al. (2015); Eban et al. (2020). Ultimately, some of these techniques Eban et al. (2020); Yvinec et al. (2021) are only introduced as a stepping stone for further compression through pruning. All these methods bear a number of limitations that render them ineffective for compression in large networks such as LLMs: first, these methods generally work at a coarse granularity level that does not allow much expressively. Second, to the best of our knowledge, there is no method that jointly learns the codebook and weight mappings in an end-to-end manner, which, we argue, is critical to

Table 1: Relative footprint of the weights $M_W$ and activations $M_A$ (in %) and total memory footprint $M_{\text{ref}}$ (in MB) of several deep neural network architectures.

| model | $M_W$ | $M_A$ | $M_{\text{ref}}$ |
|---|---|---|---|
| ResNet 18 | 94.47 | 5.53 | 87.11 |
| MobNet v2 | 95.70 | 4.30 | 121.48 |
| EffNet B0 | 95.99 | 4.01 | 129.64 |
| ViT b16 | 99.02 | 0.98 | 433.77 |

Table 2: Comparison between the latency (in ms) of models fully loaded on a CPU (i7 13th gen), GPU (A100) *v.s.* loaded on the fly from the disk of several deep neural network architectures.

| model | CPU | GPU | disk |
|---|---|---|---|
| ResNet 18 | 9.32 | 1.99 | 18.45 |
| MobNet v2 | 9.29 | 3.51 | 22.12 |
| EffNet B0 | 12.28 | 5.40 | 31.22 |
| ViT b16 | 65.90 | 3.98 | 114.75 |
| Llama 7B | 704.28 | 53.22 | 23982.74 |

enable more efficient memory compression of DNNs, particularly for LLMs. In what follows, we present the proposed JLCM method, which addresses all of these shortcomings.

## 3 METHODOLOGY

Let $F$ denote a trained neural network comprising $L$ parametric layers with weight tensors $(W_l)_{l \in \{1,\dots,L\}}$. Empirical evidence (see Table 7 from appendix A) shows that near identical performance can be achieved using the floating point 16 (fp16) format as compared to the default floating point 32 (fp32). Consequently, we note $M_{\text{ref}}$ the reference memory footprint (in Megabits) of a forward pass of $F$ using fp16. Our goal is to reduce the memory footprint $M$ of the final model $\tilde{F}$ as much as possible. We note $\alpha = \frac{M_{\text{ref}}}{M}$ the compression goal. For instance, a per-tensor quantization in 4 bits (W4) would result in $\alpha \approx 4$. More precisely, we would get $\alpha < 4$ as each scalar value is indeed stored using 4 times less memory, but an overhead occurs from the presence of extra scaling parameters. In what follows, we will design JLCM such that its only hyperparameter will be $\alpha$. Furthermore, as we focus on memory footprint during runtime, we need to evaluate the cost of each component of a deep neural network during inference.

### 3.1 MOTIVATION: DEEP NEURAL NETWORKS MEMORY FOOTPRINT

At train time, intermediate activations computed during the forward pass need to be stored for gradient backpropagation during the backward pass. Conversely, at inference time, these activations can be discarded as the subsequent layers are evaluated. Consequently, the memory footprint $M$ of a model at inference can be defined as the footprint of the weights $M_W$ and the maximum footprint of intermediate features $M_A$ to store simultaneously. The second term $M_A$ is not simply defined by the largest intermediate feature, due to the presence of skip connections in modern architectures. Nonetheless, it can be empirically computed fairly simply for any given architecture. In Table 1, we provide the contributions of the weights and activations to the memory footprint for several models. We observe that the weight systematically represent at least 94% of the memory footprint. For instance, the compression of the weight values by a factor $\alpha \approx 7.2$ of Llama 7B enables its loading on a 2017 mobile device. We argue that these results motivate the focus on weight compression for memory efficient neural networks.

In Table 2, we show the latency cost of moving tensors from different memory caches. For instance, moving the weights from disk to RAM leads to a $38.51\times$ and $450.63\times$ overhead for ViT b16 and Llama 7b, respectively. Overall, these results emphasize the importance of memory-efficient weight encoding for efficient inference. Consequently, the proposed JLCM method will enable significant inference benefits by unlocking model caching. In practice, the compression objective $\alpha$ is given by the size $M_{\text{ref}}$ of the trained neural network to run and the target hardware capacities $M_{\text{capa}}$: $\alpha > \frac{M_{\text{ref}}}{M_{\text{capa}}}$. A mobile device with 2Go RAM requires a Llama 7B to be compressed by $\alpha \approx 7.2\times$ in terms of memory footprint simply to fit on the device properly.

In what follows, we assume a given weight tensor $W \in \mathbb{R}^{n_o \times n_i}$ (*i.e.* one of the aforementioned $W_l$) with $n_o$ output neurons. Codebook-based memory compression methods suffer from two main shortcomings, namely the granularity problem and the difficulty to jointly learn the codebooks and mappings due to gradient instability.

## 3.2 SOLVING THE GRANULARITY PROBLEM WITH EFFICIENT MULTI-CODEBOOK FORMULATION

The first shortcoming of clustering techniques is that they usually assume a single distribution for all weights of the whole tensor. Formally, for a given clustering technique $\mathcal{C}$ (e.g. k-means Lloyd (1982)) we transform the weight tensor $W$ into a pair $(C, I)$ of cluster centers $C$ and mapping indices $I$ such that $W_{i,j} = \langle C; I_{i,j} \rangle$. In other words, the clustering method $\mathcal{C}$ is applied over $W$, viewed as a $n_o \times n_i$ samples of size 1. The resulting memory footprint is defined as $\Omega(C) \times 16 + \log(\Omega(C))\Omega(W)$ as compared to the original $16 \times \Omega(W)$ footprint (assuming a 16-bit representation of the codewords), with $\Omega(A)$ indicating the number of elements in a tensor $A$. As a result, the larger the codebook $C$, the higher the memory cost.

**Per-channel scaling factors:**  in order to operate at a finer granularity, we propose to draw inspiration from quantization through the use of scaling factors. Formally, instead of using a larger, common codebook, a first approach consists in sharing a smaller codebook across the weight tensor $W$ with specific per-channel scaling factors. For instance, for a per-channel granularity (one scaling factor for each row of the weight matrix), the scaling factor $s$ reads:

$$W_i = s\langle C; I_i \rangle. \tag{1}$$

This solution leads to a better control over the memory footprint $M_W = (\Omega(C) + \Omega(s)) \times 16 + \log(\Omega(C))\Omega(W)$ as we can use smaller codebooks. Intuitively, the memory reduction comes from the fact that the dominant term is $\log(\Omega(C))\Omega(W)$. However, this approach still assumes that the different neurons of a layer share similar distributions up to a scaling term.

**Weight matrix reordering:**  in order to alleviate the aforementioned shortcoming, we propose to use distinct codebooks $C_k$ rather than scaling factors. In practice, in a naïve approach, the mapping $I$ would need to encode both a codebook index (indicating to which codebook this weight shall point to) and value (the associated value in this specific codebook). Such formulation would increase the dominant term the cost $\log(\Omega(C))\Omega(W)$ in $M_W$ which, in practice, is the dominant term, as $\log(\Omega(C)) \leq 4$ and $\Omega(W) = n_i \times n_o \geq 10000$. To circumvent this limitation, we propose to systematically assign a codebook to a specific region of the weight tensor. This enables the mapping term $I$ to only encode the codebook value, as the codebook index can be derived from the row index of the considered weight. The resulting compressed weight values are defined as

$$W_{i,j} = \langle C_{\lfloor \frac{i \times k}{n_o} \rfloor}; I_{i,j} \rangle. \tag{2}$$

In practice, the number of codebooks and the number of scaling factors are directly derived from $\alpha$ (the compression goal) as follows:

$$\begin{aligned} \text{multi-scaling: } \Omega(s) &= \frac{(16 - \alpha \log(\Omega(C))) \times \Omega(W)}{16 \times \alpha} \\ \text{multi-codebooks: } k &= \frac{(16 - \alpha \log(\Omega(C))) \times \Omega(W)}{16 \times \alpha \times \Omega(C)} \end{aligned} \tag{3}$$

In order to set the codebook size $\Omega(C)$, we propose to simply use $2^{\lfloor \frac{16}{\alpha} \rfloor}$. As a result, the proposed coding can leverage the different distributions from different parts of the weight tensor (depending on the granularity), assuming that two closely related neurons (as defined by their row indexes) will have similar weight distributions. This is however not the case in practice, e.g. two neurons far from each other in the weight matrix can be similarly distributed. In order to leverage these similarities, we propose to re-order the neurons beforehand: formally, in the absence of a subsequent skip-connection, we can re-order the output neurons of a layer by re-ordering the input columns of the subsequent layer, as in Chen et al. (2021); Liang et al. (2018); Pool & Yu (2021). The re-ordering $(\sigma_i)_i$ is given by a $\mathcal{C}$ clustering of the neurons, defined by their corresponding weight values. Formally, we apply $\mathcal{C}$ to $W$ viewed as $n_o$ samples of size $n_i$. The given clusters are ordered such that neurons clustered together are contiguous in memory. We update equation 2, and get the final compressed weight values

$$W_i = \langle C_{\lfloor \frac{\sigma_i \times k}{n_o} \rfloor}; I_i \rangle. \tag{4}$$

As a result, the initialization procedure reads as follows: first, we cluster the neurons and re-order them, second cluster the scalar weight values based on the hyperparameter $\alpha$ into several codebooks or scalings. These steps are summarized in Algorithm 1 at lines 2 to 4. Stemming from this initialization, we propose to optimize both the codebooks and mappings through gradient descent.

### 3.3 JOINTLY LEARNING CODEBOOK AND MAPPINGS THROUGH GRADIENT DESCENT

In order to jointly optimize the codebook values and mappings, we adapt the sequential, layer per layer optimization introduced in Nagel et al. (2020). For a given layer, we learn the two parameters $C$ and $I$ such that they minimize the similarity loss

$$\left\|\tilde{f}(\tilde{X}) - f(X)\right\|_2^2 = \left\|\sigma((C \times \mathrm{softmax}(I))\tilde{X}) - \sigma(WX)\right\|_2^2. \tag{5}$$

In this formulation, for each weight in $W$, $\mathrm{softmax}(I)$ denote a soft assignment to one of the codebook values. In order to fight overfitting (which, given the small size of the calibration set, is a major problem in GPTQ methods Hubara et al. (2021)) while driving this mapping to a hard assignment, we propose two regularization terms: the first one, $\mathcal{L}_1(C, I) = \|C \times \mathrm{softmax}(I) - W\|_2^2$. Intuitively, $\mathcal{L}_1$ encourages the final weight tensor to be similar to the original weight values. This is a different solution to achieve a similar behavior as in AdaRound where the optimized value is bounded by the quantized space step-size. The second one, specifically enforcing hard assignment, is defined as $\mathcal{L}_2(I) = 1 - |2 \times \mathrm{softmax}(I) - 1|^\beta$. The resulting total objective $\mathcal{L}$ reads

$$\mathcal{L}(C, I) = \left\|\tilde{f}(\tilde{X}) - Y\right\|_2^2 + \mathcal{L}_1(C, I) + \lambda \mathcal{L}_2(I) \tag{6}$$

In order to keep the solution simple, we only use a single hyperparameter $\lambda$ to control the mapping sharpness. Such approach is standard in post-training optimizations Nagel et al. (2020); Li et al. (2021); Wei et al. (2022); Liu et al. (2023) and we use the same scheduler as proposed in the work of Nagel et al. (2020). Note however that, practically speaking, in this state, the optimization fails. In what follows, we delve into shortcomings of the gradient computation and propose an effective solution for joint learning the codebooks and mappings.

### 3.4 ENABLING PROXIMAL SEARCH USING AN ALTERNATIVE GRADIENT TERM

During the backward pass, we compute the gradient updates, as

$$\begin{cases} \frac{\partial \|\tilde{f}(\tilde{X}) - f(X)\|_2^2}{\partial C} = \frac{\partial \|\tilde{f}(\tilde{X}) - f(X)\|_2^2}{\partial(C \times \mathrm{softmax}(I))\tilde{X}} \times \mathrm{softmax}(I)\tilde{X} \\ \frac{\partial \|\tilde{f}(\tilde{X}) - f(X)\|_2^2}{\partial I} = \frac{\partial \|\tilde{f}(\tilde{X}) - f(X)\|_2^2}{\partial(C \times \mathrm{softmax}(I))\tilde{X}} \times \tilde{X}C \frac{\partial \mathrm{softmax}(I)}{\partial I} \end{cases} \tag{7}$$

The first gradient update in equation 7 is well suited for optimization. Intuitively, It corresponds to the grouping of the standard weight update through stochastic gradient descent filtered by the soft mapping $\mathrm{softmax}(I)$. However, on the flip side, we can see that the gradient update term in the second row of equation 7 is not adapted to the task. As $I$ is learned, the goal is to assign a new index in the codebook $C$ to a specific weight value. In other words, a large magnitude of the gradient $\frac{\partial \|\tilde{f}(\tilde{X}) - Y\|_2^2}{\partial(C \times \mathrm{softmax}(I))\tilde{X}} \times \tilde{X}$ implies that a change may be required. However, in the standard chain rule, this term is multiplied by the values of the codebook $C$, which leads to an update that is proportional to the codebook. This favors larger values in the codebook, regardless of the current indexing. For example, if we have an initial value of $0.12$ for a weight and a ternary codebook $C = (-1.23 \quad -0.09 \quad 0.17 \quad 0.96)$, with the corresponding mapping $I = (0 \quad 1 \quad 0 \quad 0)$. Then, for a negative gradient, it is likely that the indexing should shift from the second and to the third value. In practice, the chain rule would favor the update of the index corresponding to the value $0.96$ (illustrated in Figure 1). This is due to the fact that the gradient update term is an increasing function of the distance between two codebook values $C_j - C_{j'}$. To allow for a proximal behavior, we argue that we should instead use a *decreasing* function of this quantity.

We draw inspiration from straight-through estimation (STE), a method that was originally proposed for DNN quantization. In this context, STE consists in redefining a custom gradient update to sidestep the zero gradients of the rounding function. In this vein, ideally, we would want the gradient update to consider the proximity of the codewords. Consider the following function $D$:

$$D_{i,j} = \frac{\mathrm{sign}(C_{\mathrm{argmax}(I_i)} - C_j)}{1 + |C_{\mathrm{argmax}(I_i)} - C_j|} - \left(I_{\Omega(C)}\right)_{\mathrm{argmax}(I_i), j}, \tag{8}$$

where $I_{|C|}$ is the identity matrix, with the size of the codebook $C$ and the convention that $\text{sign}(0) = 1$, with index $i$ for the weight value and $j$ for the codebook value, respectively. With that convention, we get the following custom gradient updates

$$\begin{cases} \frac{\partial\left\|\tilde{f}(\tilde{X})-Y\right\|_2^2}{\partial C} = \frac{\partial\left\|\tilde{f}(\tilde{X})-Y\right\|_2^2}{\partial(C\times\text{softmax}(I))\tilde{X}} \times \text{softmax}(I)\tilde{X} \\ \frac{\partial\left\|\tilde{f}(\tilde{X})-Y\right\|_2^2}{\partial I} = \frac{\partial\left\|\tilde{f}(\tilde{X})-Y\right\|_2^2}{\partial(C\times\text{softmax}(I))\tilde{X}} \times D\tilde{X} \end{cases} \tag{9}$$

This gradient update redefinition allows for a proximal update of the codebook values, which, in turn, enables the joint learning of the codebooks and mappings (JLCM) by optimizing Equation equation 6. The proposed approach is summarized in Algorithm 1 (in Appendix B). In the following section, we provide an empirical validation of this method, as well as a comparison with existing baselines.

## 4 EXPERIMENTS

In order to evaluate the proposed JLCM method, we considered both computer vision and natural language processing tasks and models. All details are available in Appendix C. First, we compare multiple methods (clustering method, multiple scaling factors or codebooks) for the initialization, as detailed in Section 3.2. Second, we evaluate each component of the proposed JLCM method through an ablation study. Third, we compare the proposed method to the current state-of-the-art DNN memory compression techniques.

### 4.1 INITIALIZATION

Table 3: Evaluation of the clustering initialization method for the codebooks with a compression goal of $\alpha = 3.9$ and codebooks of size 16.

| | Res18 | ViT b16 | Eff B0 | Mob V2 | Res18 | ViT b16 | Eff B0 | Mob V2 |
|---|---|---|---|---|---|---|---|---|
| | Multi-Scales | | | | Multi-Codebooks | | | |
| random | 0.100 | 0.100 | 0.100 | 0.100 | 0.100 | 0.100 | 0.100 | 0.100 |
| K-means | 64.641 | 80.572 | 50.174 | **9.277** | 62.536 | 80.622 | **46.165** | 20.222 |
| B. K-means | 28.297 | 78.879 | 42.582 | 0.220 | 4.005 | 75.272 | 40.545 | 0.630 |
| Graph | 38.328 | 72.903 | 7.965 | 0.098 | 0.100 | 0.144 | 0.100 | 0.112 |
| Hierarchical | **64.937** | **80.624** | **51.773** | 8.599 | **64.277** | **80.670** | 44.334 | **22.961** |

The proposed initialization has three hyperparameters: the compression goal $\alpha$, the clustering technique $\mathcal{C}$ and the choice of using multiple scales or multiple codebooks. The *de facto* standard centroid-based method $\mathcal{C}$ is the K-means clustering Lloyd (1982) and its iterations, such as bisecting k-means Wang et al. (1998). We also considered graph-based clustering Shi & Malik (2000) and hierarchical clustering Müllner (2011). In Table 3, we report our results on ImageNet models. Our observations are two-fold. First, the clustering method plays a crucial role, as random clustering leads to near zero accuracy for all the tested networks. Second, the hierarchical and k-means clustering achieve the highest performance, which can be explained by the fact that, for scalar clustering, they tend to behave similarly. Still, it appears the hierarchical clustering leads to a slightly higher performance over the tested configurations. Consequently, we will use this initialization scheme for the upcoming experiments. Furthermore, it appears that the proposed multi-scaling approach performs best on convolutional layers. On the flip side, the multi-hashlists variants are more effective on fully-connected layers. Intuitively, we suggest that relative scales are more important for convolutional 2D kernels while scalar values nuance is more important for fully-connected layers. Consequently, for the remainder of the article, JLCM will use multi-scalings for CNNs architectures and multi-hashlists for transformers.

### 4.2 ABLATION STUDY

In Table 4, we evaluate the influence of each component of the optimization phase in JLCM. More specifically, *learn C* means that we only update the codebook during optimization by applying the

Table 4: Ablation of the optimization components of JLCM for a compression rate $\alpha = 7.5$.

| learn $C$ | learn $C$ & $I$ | Proximal search | acc for ResNet 18 | acc for ViT |
|:---:|:---:|:---:|:---:|:---:|
| ✗ | ✗ | ✗ | 20.555 | 60.691 |
| ✓ | ✗ | ✗ | 22.149 | 63.912 |
| ✓ | ✓ | ✗ | 20.343 | 57.800 |
| ✓ | ✓ | ✓ | **41.779** | **69.996** |

Table 5: Comparison to state-of-the-art compression techniques on ImageNet.

| architecture | method | Compression rate ($\alpha$) | optimization | integer | acc |
|:---:|:---:|:---:|:---:|:---:|:---:|
| | DFQ | 5.33 (W3/A16) | ✗ | ✓ | 0.442 |
| | SQuant | 5.33 (W3/A16) | ✗ | ✓ | 19.532 |
| | PowerQuant | 5.33 (W3/A16) | ✗ | ✓ | 3.909 |
| | RED++ | 5.33 | ✗ | ✗ | 14.124 |
| | JLCM (init) | 5.33 | ✗ | ✗ | **41.133** |
| Resnet 18 | JLCM (init) | 7.5 | ✗ | ✗ | 20.555 |
| | AdaRound | 5.33 (W3/A16) | ✓ | ✓ | 19.634 |
| | BrecQ | 5.33 (W3/A16) | ✓ | ✓ | 52.943 |
| | NUPES | 5.33 (W3/A16) | ✓ | ✗ | 53.000 |
| | JLCM | 5.33 | ✓ | ✗ | **62.939** |
| | JLCM | 7.5 | ✓ | ✗ | 41.779 |
| | DFQ | 5.33 (W3/A16) | ✗ | ✓ | 11.678 |
| | SQuant | 5.33 (W3/A16) | ✗ | ✓ | 13.804 |
| | PowerQuant | 5.33 (W3/A16) | ✗ | ✓ | 75.330 |
| | RED++ | 5.33 | ✗ | ✗ | 75.743 |
| | JLCM (init) | 5.33 | ✗ | ✗ | **76.316** |
| ViT b16 | JLCM (init) | 7.5 | ✗ | ✗ | 60.691 |
| | AdaRound | 5.33 (W3/A16) | ✓ | ✓ | 61.154 |
| | BrecQ | 5.33 (W3/A16) | ✓ | ✓ | 73.123 |
| | NUPES | 5.33 (W3/A16) | ✓ | ✗ | 77.231 |
| | JLCM | 5.33 | ✓ | ✗ | **80.558** |
| | JLCM | 7.5 | ✓ | ✗ | 69.996 |

first row update in Equation equation 7. This leads to a steady performance improvement. Furthermore, jointly optimizing the codebook and mappings (*learn $C$ & $I$*) by applying the whole equation 7 leads to a decrease in performance: this is due to the limitations of the naive gradient update pinpointed in Section 3.4. Nevertheless, joint learning can be managed by using the proposed alternative gradient update term (Proximal search, equation 8), which dramatically enhances the accuracy of the compressed networks in both configurations. In what follows, we show that this framework allows to significantly outperform existing methods for memory compression of DNNs.

## 4.3 COMPARISON TO STATE-OF-THE-ART METHODS

In Table 5, we report a comparison of JLCM to state-of-the-art post-training compression techniques on ImageNet. We distinguished convolutional neural networks (ResNet 18) and vision transformers (ViT) as well as the data usage (data-free (white) *v.s.* data-driven (light gray) methods) and report results at two different compression goals, 5.33 (float 16 → int3) and 7.5 (float 16 → ≈ int2). On CNNs, it appears that the data-free JLCM initialization already vastly outperforms other data-free methods Nagel et al. (2019); Cong et al. (2022); Yvinec et al. (2022; 2023c) as well as AdaRound Nagel et al. (2020) at an equal compression goal. This result is maintained in the data-driven context, where JLCM improves the previous state-of-the-art by 9.939 points. On ViT b16, PowerQuant and RED++ already achieve strong results: consequently, improving over these results is more chal-

Table 6: Evaluation on Generative AI models.

(a) CLIP Score of a stable diffusion model (original score: 28.5353) 2.0 on DiffusionDB.

| method | compression $\alpha$ | acc |
|---|---|---|
| DFQ | 3.9 (W4/A16) | 25.369 |
| SQuant | 3.9 (W4/A16) | 26.237 |
| PQuant | 3.9 (W4/A16) | 27.510 |
| RED++ | 3.9 | 27.498 |
| JLCM (init) | 3.9 | **28.471** |

(b) Average common sense reasoning performance of a Llama 7B (original score: 56.140).

| method | compression $\alpha$ | acc |
|---|---|---|
| SQuant | 5.33 (W3/A16) | 32.725 |
| LLM.int8() | 2.14 (W8/A8) | 55.208 |
| PQuant | 5.33 (W3/A16) | 33.466 |
| RED++ | 5.33 | 36.430 |
| OPTQ | 5.33 (W3/A16) | 49.938 |
| AWQ | 5.8 | 47.678 |
| SqueezeLLM | 5.8 | 48.169 |
| JLCM (init) | 3.9 | 56.153 |
| JLCM (init) | 5.33 | 45.314 |
| JLCM | 5.33 | **55.940** |
| JLCM | 7.0 | **53.081** |
| JLCM (+ REx) | 6.95 | **56.002** |

lenging. Still, JLCM manages to reach over 99% of the original model accuracy in its data-driven variant. This corresponds to a 3.327 points improvement over Yvinec et al. (2023b).

In Table 6a and 6b, we report our results on larger, more recent generative models. Our results show that JLCM offers a significant improvement in the data-free set-up. For instance, on the Stable Diffusion model, JLCM preserves 99.8% of the original performance, which is 3.4% higher than the previous state-of-the-art method (RED++). Similarly, on Llama 7B, we observe an 8.884% improvement. This can be attributed to the finer-grained weight compression, which enables JLCM to better address the challenge of outlying values encoding, as discussed by Dettmers et al. (2022). In the data-driven setup, JLCM further increase the gap with previous techniques, which struggle to scale to large models. Furthermore, on Llama, JLCM reaches a 3.143% higher average score on common sense reasoning tasks as compared to OPTQ Frantar et al. (2022) with a 31.33% higher compression rate. In addition, as compared to SqueezeLLM Kim et al. (2023), JLCM not only achieves higher fidelity but at higher compression rates which can be attributed to our more effective usage of codebooks and proximal learning. On top of this, when JLCM is combined with outlier specific compression from REX Yvinec et al. (2023a) (also used by SQueezeLLM), we reach the original performance at similar compression rate, which is a major improvement over existing methods. Last but not least, we also provide qualitative results on diffusion models to complete this score-based evaluation (see Appendix D).

## 5 CONCLUSION

In this work, we pinpointed two major drawbacks of existing DNN memory compression techniques: the granularity (*i.e.* the ability to use multiple codebooks per tensor with no mapping overhead) as well as the difficulty to optimize the mapping through stochastic gradient descent, as the process favors extreme codebook values over proximal ones. To circumvent these limitations, we introduced a novel efficient codebook mapping which leverages tensor structure. To do so, we proposed to re-order neurons in order to group them by distribution similarity. We then proposed to either apply different scaling factors (which, experimentally, works best for convolutional layers), or encode each group using a separate codebook (which is more efficient for dense layers, e.g. in transformers), removing the need for codebook indexing that causes memory overhead on the value mapping. We then propose a joint learning of the codebooks and weight mappings (JLCM) method that draws inspiration from gradient-based post-training quantization techniques. Last but not least, we propose an alternative gradient update rule that allows a proximal search within codebook values. The proposed method significantly outperforms other approaches for memory-efficient inference of deep neural networks. As an example, JLCM compresses a Llama-7B model to fit on a 2Go device while retaining 95% of its original performance, which represents a massive improvement over previous state-of-the-art, and an important stepping stone for the deployment of ever-growing deep neural networks.

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
