Table 7: Evaluation of classification models using fp32/ fp16 inference.

| model | acc in fp32 | acc in fp16 |
|---|---|---|
| ResNet 18 | 68.632 | 68.632 |
| MobileNet v2 | 71.199 | 71.199 |
| EfficientNet B0 | 76.402 | 76.402 |
| ViT b16 | 80.966 | 80.966 |

---

**Algorithm 1** JLCM Algorithm

---

**Require:** pre-trained model $F$ with layers $f_l$, compression goal $\alpha$ and clustering technique $\mathcal{C}$

1: **for** $l \in \{1, \dots, L\}$ **do**
2:     $\_, \sigma \leftarrow \mathcal{C}(W_l)$                                                      ▷ $W_l$ seen as $n_o$ samples of size $n_i$.
3:     re-order $W_l$ and $W_{l+1}$ with $\sigma$
4:     $C, I \leftarrow \mathcal{C}(W_l)$                                                       ▷ $W_l$ seen as $n_o \times n_i$ scalar samples.
5:     **for** $i \in \{1, \dots, 10000\}$ **do**
6:         get $\tilde{X}, X$ intermediate features from the hashed and original models
7:         minimize equation 6
8:     **end for**
9: **end for**

---

## A    FP16 AND FP32

In Table 7, we report a comparison between floating point 16 (fp16) and floating point 32 (fp32). These results confirm that fp16 is a more relevant baseline for modern efficient inference.

## B    JLCM ALGORITHM

We provide a pseudocode algorithm for the proposed method in Algorithm 1.

## C    IMPLEMENTATION DETAILS

Regarding computer vision, we worked on convolutional neural networks, namely ResNet 18 He et al. (2016), MobileNet v2 Sandler et al. (2018) and EfficientNet B0 Tan & Le (2019) as well as the ViT b16 transformer architecture Dosovitskiy et al. (2021), all trained for ImageNet Deng et al. (2009). To complete our study of vision models, we evaluated the impact of JLCM on a diffusion model Rombach et al. (2021) which is evaluated using the CLIP score Hessel et al. (2021) on prompts from DiffusionDB Wang et al. (2022). For NLP tasks, we focused our efforts on Llama 7b Touvron et al. (2023) which we evaluate on common sense reasoning benchmarks Clark et al. (2019); Bisk et al. (2020); Zellers et al. (2019); Sakaguchi et al. (2021); Clark et al. (2018); Mihaylov et al. (2018). For our experiments, we downloaded the pre-trained weight values from Torchvision when possible and from HuggingFace otherwise. We used a single A100 GPU with a torch implementation for all of our experiments with a fixed seed of 12345. Unless otherwise stated, we applied the default hyperparameter values for the optimizers and clustering techniques, which come from the Scikit-learn package.

# D    DIFFUSION MODELS VISUALIZATION

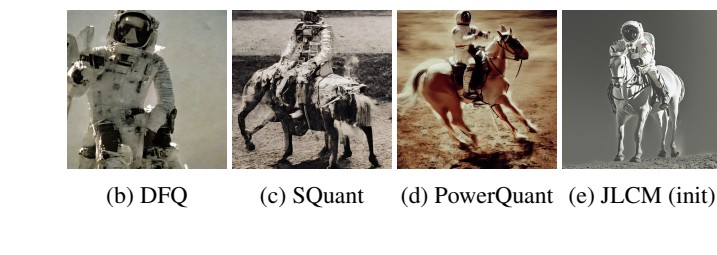

(b) DFQ          (c) SQuant          (d) PowerQuant    (e) JLCM (init)

(a) original model

Figure 2: Input prompt: "a photo of an astronaut riding a horse"

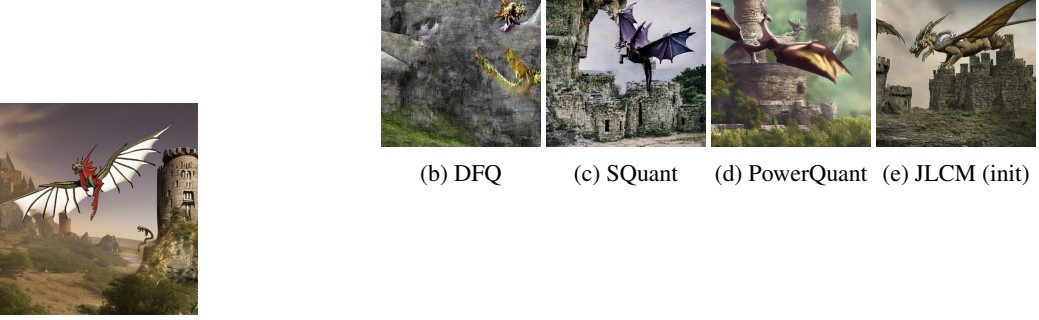

(b) DFQ          (c) SQuant          (d) PowerQuant    (e) JLCM (init)

(a) original model

Figure 3: Input prompt: "a photo of a dragon flying next to a big stone castle"

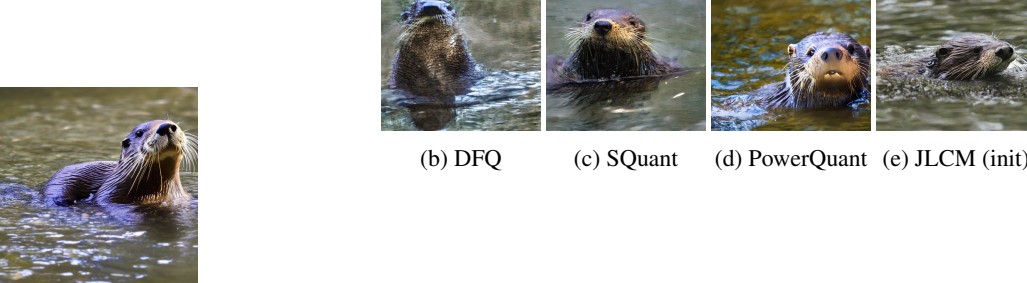

(b) DFQ          (c) SQuant          (d) PowerQuant    (e) JLCM (init)

(a) original model

Figure 4: Input prompt: "a photo of an otter swimming in a river"