# OpenReview forum: "Network Memory Footprint Compression Through Jointly Learnable Codebooks and Mappings"
_ICLR.cc/2024/Conference — ICLR 2024 poster_

### Official Review · Reviewer_fJGw · 2023-10-30

**Soundness:** 2 fair
**Presentation:** 2 fair
**Contribution:** 2 fair
**Rating:** 6
**Confidence:** 2

**Summary:**

The novel ALAM framework introduced in this paper leverages average quantization and a simple sensitivity calculation method to reduce memory usage in LLMs without affecting their training efficacy. This approach minimizes gradient variance by compressing activations to their group average values, allowing for effective compression to less than 1 bit. Additionally, ALAM employs a novel sensitivity calculation that uses the L2 norm of parameter gradients, greatly reducing memory overhead. In testing, ALAM achieves up to a 12.5× compression rate for activation memory in LLMs without sacrificing accuracy.

**Strengths:**

1.The paper applied a re-order method on neurons to reduce the memory overhead, which is quite novel.

2.This paper jointly optimized the mapping and codebooks, and the proximal search method can be used to the modified gradient update method.

3.The evaluation demonstrates the effectiveness.

**Weaknesses:**

1.It is hard to understanding the background part, especially the challenges of conventional approach. It would be better to provide a straightforward illustration with a figure or algorithm.
2.Some typos. In the beginning of sec3.2, “the” should be “The“; “ram” in sec 3.1 should be “RAM”; Something is missing in Equation 3.
3.The metric in evaluation part is accuracy and compression ratio. Can the proposed idea bring more benefits like accelerating the inference/training/finetuning?

**Questions:**

1.It would be better to explain why the second term $log(\Omega(C))\Omega(W)$dominates. It seems a common sense but I cannot see it in the context.

2.Can you provide some evaluation results in inference or training or finetuning performance in efficiency (throughput or energy efficiency)?

---

> ### Author Response · Authors · 2023-11-13
> **response to reviewer fJGw**
>
> ### It is hard to understanding the background part, especially the challenges of conventional approach. It would be better to provide a straightforward illustration with a figure or algorithm.
>
> We introduced a novel figure to highlight these elements. First, we illustrate the challenges of the baseline approach and second, our contributions both in terms of multiple codebooks/scalings and gradient updates.
>
> ### Some typos. In the beginning of sec3.2, “the” should be “The“; “ram” in sec 3.1 should be “RAM”; Something is missing in Equation 3.
>
> Thank you for pointing these out. We fixed these typos in the revision version and doubled checked the remainder of the article.
>
> ### The metric in evaluation part is accuracy and compression ratio. Can the proposed idea bring more benefits like accelerating the inference/training/finetuning?
>
> the primary aim of the proposed JLCM method is to preserve the accuracy of the model while significantly reducing its memory footprint. However, as illustrated in Table 2, memory limitations can force the weights to be loaded on-the-fly for inference, leading to severe computational burden: this is especially true on large models (see last row in Table 2 for instance), as the architectures become more and more parameter hungry.
>
> In this context, using JLCM can translate in huge latency improvements at inference time, e.g. on low-power or low-memory hardware where the original model cannot fit entirely, or if we consider very large models (e.g. LLMs).
>
> Secondly, to answer your question, and although not studied in the present paper, one could decide to only optimize the codebooks during a post-hoc fine-tuning phase. In this case ,the proposed method would behave similarly to a foldeable adapter (e.g. LoRa [1], which can allow to finetune or fewer examples or using less memory), which could be an interesting lead for future work.
>
> ### It would be better to explain why the second term $\log(\Omega(C))\Omega(W)$ dominates. It seems a common sense but I cannot see it in the context.
>
> To answer you concern, we clarified this point in the revised version of the paper. In short, for a dense layer of weights $W \in \mathbb{R}^{n_i \times n_o}$, we have $\Omega(W) = n_i \times n_o$ which is often larger than $100000$ and can grow to several millions in some LLMs. On the other end, $\log(\Omega(C))$ corresponds to the size of our codebooks which is empirically bounded by $\Omega(C) \leq 16$, i.e. we have $\log(\Omega(C)) \leq 4 \ll 100000 \leq \Omega(W)$.
>
> ### Can you provide some evaluation results in inference or training or finetuning performance in efficiency (throughput or energy efficiency)?
>
> In terms of latency, we refer to Table 2 in our original submission (see also the answer to your previous question, in short JLCM has the potential to increase the latency in the cases where the model is too big to be loaded on device at once, and memory swaps have to be done). This was shown in SQueeze LLM [2]. With respect to throughput, we share these new results
>
> | model | GPU (A100) | GPU (A100) with offload on disk |
> | :---: | :---: | :---: |
> | ViT b16  | 1333.33 | 280.70 |
> | LLama 7B | 28.57 | 1.33 |
>
> In both cases, we measure the impact of a model not fitting on the device cache (e.g. the V-RAM for a GPU) and thus needing data transfers from disk. This occurs when a model is too large to be fully loaded on the device cache (e.g. V-RAm for a GPU). In such circumstances, we need to offload some paramters on the disk and load them when necessary (this process is optimized with the accelerate library from huggingface). In this experiment, we imitate the situation where the GPU has 2GB of VRAM (simulating the memory on a smartphone). Still, having the model fully loaded leads to massive throughput benefits. This can be acheived through memory footprint compression as in JLCM.  We hope that these results further motivate the relevance of the proposed JLCM method.
>
> ### references
>
> [1] Hu, Edward J., et al. "Lora: Low-rank adaptation of large language models." arXiv preprint arXiv:2106.09685 (2021).
>
> [2] Kim, Sehoon, et al. "SqueezeLLM: Dense-and-Sparse Quantization." arXiv preprint arXiv:2306.07629 (2023).

---

### Official Review · Reviewer_P3Eg · 2023-10-30

**Soundness:** 3 good
**Presentation:** 3 good
**Contribution:** 3 good
**Rating:** 8
**Confidence:** 2

**Summary:**

The authors propose a series of improvements to codebook-based weight compression schemes for deep neural networks with the aim of enabling larger models to fit in the limited on device storage for accelerators like GPUs. Their method is based on three core changes relative to existing methods. First, they apply a neuron re-ordering to group weights with similar distributions. This allows for finer-grained application of codebooks to weights without increasing the storage overhead of the compressed representation. Second, they propose to jointly learn the codebook and codebook mappings, similar to gradient-based post-training quantization schemes. Lastly, they modify the gradient update for the codebook mappings to enable more effective optimization.

**Strengths:**

Despite my lack of experience with the topic of this paper I found the text reasonably easy to follow. I think the paper is well written and the methods appear to be sound to me. In particular, the impact of part 3 of the proposed method (improved gradient estimator) seems to be considerable based on the ablation results presented in Table 4.

**Weaknesses:**

I am not an expert on compression of neural network weights for storage optimization but I did not identify any particular weaknesses in the methodology. The technique appears to be reasonable and the results appear to be sound based on my review of the paper.

**Questions:**

The neuron permutations you use in your method remind me of the channel permutations that are used by N:M sparsification methods [1]. Drawing this connection could be interesting in your related work section.

In a number of places you use “Go” as a unit - is this intentional? Was “Go” supposed to be “GB”?

In Table 2, I’m curious to understand how you implemented offloading of parameters to disk. Are transfers from disk to GPU pipelined with computation to hide as much transfer latency as possible? What batch sizes were used for each model?

[1] https://proceedings.neurips.cc/paper_files/paper/2021/hash/6e8404c3b93a9527c8db241a1846599a-Abstract.html

---

> ### Author Response · Authors · 2023-11-13
> **response to reviewer P3Eg**
>
> We thank the reviewer for their interest in the method and we appreciate the comment on readibility, especially provided this is not their exact domain of expertise.
>
> ### The neuron permutations you use in your method remind me of the channel permutations that are used by N:M sparsification methods [1]. Drawing this connection could be interesting in your related work section.
>
> We agree that our paper would benefit from this and other references on weight permutation. They were added directly in the corresponding section in the methodology.
>
> ### In a number of places you use “Go” as a unit - is this intentional? Was “Go” supposed to be “GB”?
>
> This was indeed a typo that was corrected in the revised version.
>
> ### In Table 2, I’m curious to understand how you implemented offloading of parameters to disk. Are transfers from disk to GPU pipelined with computation to hide as much transfer latency as possible? What batch sizes were used for each model?
>
> The implementation of this offloading can be achieved using the accelerate library from huggingface, from our experiments it is well optimized in order to hide as much as possible the cost of data transfers.
>
> Regarding the batch-size, we measure the latency which is defined using a batch size of 1 (table 2). On the flip side, if we want to maximize the throughput (number of predictions per second with optimal batch size = 32), we observe very similar results (especially on larger models). In this experiment, we imitate the situation where the GPU has 2GB of VRAM (simulating the memory on a smartphone).
>
> | model | GPU (A100) | GPU (A100) with offload on disk |
> | :---: | :---: | :---: |
> | ViT b16  | 1333.33 | 280.70 |
> | LLama 7B | 28.57 | 1.33 |
>
> We hope that these answers address your questions.

---

### Official Review · Reviewer_kEDS · 2023-11-01

**Soundness:** 2 fair
**Presentation:** 2 fair
**Contribution:** 2 fair
**Rating:** 5
**Confidence:** 4

**Summary:**

This paper compresses memory footprint in deep neural networks inference by a codebook-based approach, mainly solving the granularity problem and the training problem suffered by the previous works. This paper tried two methods to solve the granularity problem, setting per-channel scaling factors with one codebook or using multiple codebooks with weight matrix reordering. To enable proximal search thus solving the training problem, this method uses custom gradient updates inspired by STE.

**Strengths:**

1. In the experiment section, this method shows advantages over baselines.

2. This paper provides a detailed analysis of memory usage in different settings.

**Weaknesses:**

The paper has some unclear statements that require more explanation:

1.  The settings for the number of codebooks and the number of scaling factors should be further explained. Although the authors mention in the paper that these values are determined by compression goal, the process needs more details and therelationship with the weight distribution needs to be explained as well.

2. This paper does not explain how to cluster the neurons. There is also no data to prove that “two neurons far from each other in the weight matrix can be similarly distributed”.

3. The meaning of the X-axis in Figure 1 is confusing. For different curves, the X-axis seems to have different meanings. These should be marked on the figure.

4. For the LLM experiments in Table 6, how are the results compared with more recent works, e.g., SqueezeLLM, AWQ, etc.

**Questions:**

Please refer to the weaknesses.

---

> ### Author Response · Authors · 2023-11-13
> **Response to reviewer kEDS**
>
> We would like to thank you for your interest in the proposed method and for highlighting the empricial benefits offered by the proposed approach. In the following, we share our responses to your concerns.
>
> ### The settings for the number of codebooks and the number of scaling factors should be further explained. Although the authors mention in the paper that these values are determined by compression goal, the process needs more details and therelationship with the weight distribution needs to be explained as well.
>
> In our original submission, we define the memory footprint of a weight tensor $W \in \mathbb{R}^{n_i \times n_o}$ based on the cost of encoding a single scalar value (in 16 bits floating point) mutliplied by the number of weights (end of introduction of section 3.2), i.e.
> $$
> M = \Omega(W) \times 16 = n_i \times n_o \times 16
> $$
> Now assuming that we know the target hardware capacities (in terms of memory) $M_{\text{capa}}$ and the original memory footprint of the original model $M_{\text{ref}}$, we can derive the target compression goal $\alpha\geq\frac{M_{\text{ref}}}{M_{\text{capa}}}$. Now, we have two cases: multi-codebooks and multi-scalings. In our original submission, we define the memory cost for both as follows
> $$
> \begin{cases}
> \text{multi-codebooks:} & \Omega(W) \times \log(\Omega(C)) + k \times \Omega(C) \times 16 \\
> \text{multi-scalings:} & \Omega(W) \times \log(\Omega(C)) + (\Omega(s) + \Omega(C)) \times 16
> \end{cases}
> $$
> where $k$ is the number of codebooks and $\Omega(s)$ the number of scaling terms. If we combine this equation with the constraint of $\alpha$, we can derive the number of codebooks and scaling terms immediately (as in equation 3 of the original paper)
> $$
> \begin{cases}
> \text{multi-codebooks:} & \Omega(W) \times 16 = \alpha \left(\Omega(W) \times \log(\Omega(C)) + k \times \Omega(C) \times 16\right) \\
> \text{multi-scalings:} & \Omega(W) \times 16 = \alpha \left(\Omega(W) \times \log(\Omega(C))+ (\Omega(s) + \Omega(C)) \times 16\right)
> \end{cases}
> $$
> The multi-codebook encoding is more robust to weight tensor with a lot of variation per-neuron in their distribution. On the flip side, multi-scaling encoding is more robust to weight tensors with signifant shifts in their support range but low variation in their distribution.
>
> We aknowledge that our paper would benefit from these details and we added them to our rebuttal revision.
>
>
> ### This paper does not explain how to cluster the neurons. There is also no data to prove that “two neurons far from each other in the weight matrix can be similarly distributed”.
>
> In order to cluster neurons, we use the same clsutering technique $\mathcal{C}$ for the weights and the neurons. This information was missing above equation 4. We added it in the revised article. In order to re-order the output neurons of a layer, one simply needs to re-order the input neurons of the subsequent layers. We added references which detail this process which has been widely adopted in the literature \[1,2,3\]. Similarly, the fact that two neurons of the same layer can be similalry distributed and be "far from each other" (i.e. in term of indexes) was illustrated in DFQ [4]. For the sake of having a self-contained paper, we added these elements of clarification in the revised manuscript.
>
> ### The meaning of the X-axis in Figure 1 is confusing. For different curves, the X-axis seems to have different meanings. These should be marked on the figure.
>
> This figure was indeed confusing, we updated it for a novel one which illustrates both our main contributions at once. Please refer to the revised article.

---

> > ### Author Response · Authors · 2023-11-13
> > **remainder**
> >
> > ### For the LLM experiments in Table 6, how are the results compared with more recent works, e.g., SqueezeLLM, AWQ, etc.
> >
> > As requested, we added performance of AWQ and Squeeze LLM to our initial comparison to the state-of-the-art with Llama 7B on common sense reasoning.
> >
> > | method | compression $\alpha$ | acc |
> > | :---: | :---: | :---: |
> > | original model | - | 56.140 |
> > | OPTQ (g128) | 5.33 | 49.938 |
> > | AWQ (g128) | 5.8 | 47.678 |
> > | SqueezeLLM | 5.8 | 48.169 |
> > | SqueezeLLM | 6.3 | 42.208 |
> > | JLCM | 7.0 | **53.081** |
> > | JLCM + REx | 6.95 | **56.002** |
> >
> > We obtained these results using the publicly available implementation of [SqueezeLLM](https://github.com/SqueezeAILab/SqueezeLLM) and [AWQ](https://github.com/mit-han-lab/llm-awq). It is worth noting that SqueezeLLM leverages two elements: clustering (similar to JLCM) and outlier-specific sparse encoding (identical to REx [5]). For this reason, we added results from the proposed JLCM method in combination with the sparse encoding of the outliers. Overall, it appears that JLCM offers better trade-offs than AWQ and Squeeze LLM, which can be explained by the fact that JLCM can leverage inexpensive fine-tuning and finer-grained weight encodings through multiple codebooks. On the flip side, the usage of the hessian importance in the k-means (a contribution of the SqueezeLLM paper) seem less impactful. Finally, the addition of a separate encoding of the outlier values (JLCM + the recently proposed REx [5]) fixes the slight performance drop of JLCM in this extreme compression setup.
> >
> > We believe that these additional results show the quality of the proposed method and can fully convince the reviewer of the interest of JLCM.
> >
> > ### References
> >
> > [1] Chen, Tianyi, et al. "Only train once: A one-shot neural network training and pruning framework." Advances in Neural Information Processing Systems 34 (2021)
> >
> > [2] Liang, Ling, et al. "Crossbar-aware neural network pruning." IEEE Access 6 (2018)
> >
> > [3] Riccomagno, Martin M., and Alex L. Kolodkin. "Sculpting neural circuits by axon and dendrite pruning." Annual review of cell and developmental biology 31 (2015)
> >
> > [4] Nagel, Markus, et al. "Data-free quantization through weight equalization and bias correction." Proceedings of the IEEE/CVF International Conference on Computer Vision. 2019.
> >
> > [5] Yvinec, Edouard, et al. "REx: Data-Free Residual Quantization Error Expansion." Advances in Neural Information Processing Systems 36 (2023)

---

### Author Response · Authors · 2023-11-21
**Discussion Summary**

Dear reviewers,

We would like to thank you again for your time and efforts in providing relevant reviews to our work. In order to simplify the remainder of this discussion period, we summarize our response here:
 1. Some clarifications were requested and made in the updated version of the submission
 2. We introduced an overview of both challenges and solutions provided in JLCM (to efficient encoding of multiple codebooks and proximal search of codebooks mappings through stochastic gradient descent)
 3. As requested by one reviewer, we  highlighted how JLCM massively outperforms other codebook-based methods (including very recent work AWQ [[2](https://arxiv.org/pdf/2306.00978.pdf)] and SqueezeLLM [[3](https://arxiv.org/pdf/2306.07629.pdf)]), to further complete our initial evaluation on stable diffusion models, convnets, image transformers and LLMs.

On top of these comments, the reviewers showed interest for the ability to use the memory footprint reduction from JLCM to accelerate inference. To this end, we provided latency (response time for a single prediction) scores in the original paper. We showed that moving the weights from disk to RAM enables from 38.51$\times$ to 450.63$\times$ improvements. This can be explained by the efficiency of modern hardware at performing computations and low efficiency at making memory accesses. In the rebuttal, we extended these results to throughput (maximum number of predictions in a second), which further confirmed the great benefits of JLCM.

| method | compression $\alpha$ | acc |
| :---: | :---: | :---: |
| original model | - | 56.140 |
| OPTQ [[1](https://openreview.net/pdf?id=tcbBPnfwxS)] (g128) | 5.33 | 49.938 |
| AWQ [[2](https://arxiv.org/pdf/2306.00978.pdf)] (g128) | 5.8 | 47.678 |
| SqueezeLLM [[3](https://arxiv.org/pdf/2306.07629.pdf)] | 5.8 | 48.169 |
| SqueezeLLM | 6.3 | 42.208 |
| JLCM | 7.0 | **53.081** |
| JLCM + REx [[4](https://arxiv.org/pdf/2203.14645.pdf)] | 6.95 | **56.002** |

If you have any further remarks or questions, we will gladly discuss them.

---

### Meta-Review · Area_Chair_K5KN · 2023-12-11

**Metareview:**

This paper proposes a model compression method which jointly learn the codebook and mapping. The proposed method can achieve high compression ratio for models including llama-7b. The reviewer affirmed the effectiveness of the method. Reviewer required clarifications should be included in the final version of the paper.

**Justification For Why Not Higher Score:**

The presentation needs to be improved.

**Justification For Why Not Lower Score:**

The method is effective and the paper has contribution to the model compression community.

---

### Decision · Program_Chairs · 2024-01-16

Accept (poster)